# Genomic Identification and Expression Profiling of Lesion Simulating Disease Genes in Alfalfa (*Medicago sativa*) Elucidate Their Responsiveness to Seed Vigor

**DOI:** 10.3390/antiox12091768

**Published:** 2023-09-15

**Authors:** Shoujiang Sun, Wen Ma, Zhicheng Jia, Chengming Ou, Manli Li, Peisheng Mao

**Affiliations:** Forage Seed Laboratory, College of Grassland Science and Technology, China Agricultural University, Beijing 100193, China; shoujiangsun@cau.edu.cn (S.S.); mmw21@cau.edu.cn (W.M.); zhicheng.jia@cau.edu.cn (Z.J.); b20203240983@cau.edu.cn (C.O.); lml@cau.edu.cn (M.L.)

**Keywords:** alfalfa, seed aging, seed vigor, programmed cell death (PCD), *LSD* gene family, expression profiling

## Abstract

Seed aging, a common physiological phenomenon during forage seed storage, is a crucial factor contributing to a loss of vigor, resulting in delayed seed germination and seedling growth, as well as limiting the production of hay. Extensive bodies of research are dedicated to the study of seed aging, with a particular focus on the role of the production and accumulation of reactive oxygen species (ROS) and the ensuing oxidative damage during storage as a primary cause of decreases in seed vigor. To preserve optimal seed vigor, ROS levels must be regulated. The excessive accumulation of ROS can trigger programmed cell death (PCD), which causes the seed to lose vigor permanently. LESION SIMULATING DISEASE (LSD) is one of the proteins that regulate PCD, encodes a small C2C2 zinc finger protein, and plays a molecular function as a transcriptional regulator and scaffold protein. However, genome-wide analysis of *LSD* genes has not been performed for alfalfa (*Medicago sativa*), as one of the most important crop species, and, presently, the molecular regulation mechanism of seed aging is not clear enough. Numerous studies have also been unable to explain the essence of seed aging for *LSD* gene regulating PCD and affecting seed vigor. In this study, we obtained six *MsLSD* genes in total from the alfalfa (cultivar Zhongmu No. 1) genome. Phylogenetic analysis demonstrated that the *MsLSD* genes could be classified into three subgroups. In addition, six *MsLSD* genes were unevenly mapped on three chromosomes in alfalfa. Gene duplication analysis demonstrated that segmental duplication was the key driving force for the expansion of this gene family during evolution. Expression analysis of six *MsLSD* genes in various tissues and germinating seeds presented their different expressions. RT-qPCR analysis revealed that the expression of three *MsLSD* genes, including *MsLSD2*, *MsLSD5*, and *MsLSD6*, was significantly induced by seed aging treatment, suggesting that they might play an important role in maintaining seed vigor. Although this finding will provide valuable insights into unveiling the molecular mechanism involved in losing vigor and new strategies to improve alfalfa seed germinability, additional research must comprehensively elucidate the precise pathways through which the *MsLSD* genes regulate seed vigor.

## 1. Introduction

Seed quality has a pivotal influence on agricultural production, the efficient conservation of genetic resources, and safeguarding biodiversity [1]. Losing seed vigor over time, known as seed aging, constitutes a common physiological phenomenon during seed storage, resulting from either natural or accelerated aging processes. Seed aging is closely associated with various intricate cellular processes, including mitochondrial alterations, programmed cell death (PCD), DNA repair, antioxidant defense mechanisms, telomere length, and epigenetic regulation [2]. Furthermore, the repercussions of seed aging extend to seedling growth, plant yield, and overall quality during subsequent stages of development. Thus, investigating the underlying mechanisms governing seed aging holds paramount significance. Extensive research has been dedicated to the study of seed aging, with a particular focus on the role of the production and accumulation of reactive oxygen species (ROS) and the ensuing oxidative damage during storage, as a primary cause of decreases in seed vigor [3]. To preserve optimal seed vigor, ROS levels must be regulated through enzymatic and non-enzymatic mechanisms, and DNA repair processes be facilitated within the seed [4]. Nonetheless, an in-depth comprehension of the precise functions of ROS in determining seed longevity and the aging process is imperative. Recent studies have even unveiled that seed aging may be instigated via PCD, introducing new dimensions to this intricate phenomenon.

PCD is a fundamental process crucial for the maintenance of tissue homeostasis, the elimination of undesirable or damaged cells, and the orchestration of proper developmental pathways triggered by intrinsic and extrinsic environmental stimuli [5]. PCD has been well studied in animals, though scientific knowledge is limited regarding the mechanisms that regulate and execute plant cell death [6]. Hydrogen peroxide (H_2_O_2_) and other ROS are recognized as key modulators of PCD and many other biological processes, including growth, development, and stress adaptation [7]. Notably, the research on elm (*Ulmus pumila*) aged seeds has revealed a systematic and progressive occurrence of PCD (Figure 1). The controlled deterioration treatment (CDT) of seeds triggers the release of cytochrome c (Cyt c). It causes an increase in caspase-3-like/DEVDase activity, both of which can be suppressed by ascorbic acid (AsA) and the caspase-3 inhibitor Ac-DEVD-CHO, respectively [8]. During the process of seed aging, there is a concomitant elevation of cytosolic calcium levels, which coincides with the release of Cyt c facilitated by the activation of the permeability transition pore (PTP) through the adenine nucleotide translocator (ANT) and voltage-dependent anion channel (VDAC). Subsequently, the liberated Cyt c instigates the activation of caspases and triggers the downstream proteolytic cascades involved in the execution of PCD [8]. The in situ localization of ROS production during CDT reveals distinct spatial–temporal patterns of ROS that correlate with the characteristic features of PCD. Notably, multiple antioxidant elements are initially activated during the early stages of CDT but are subsequently depleted as PCD progresses (Figure 1). Researchers identified PCD as a crucial asymmetric mechanism during elm seed CDT and proposed its significant role in seed deterioration [8].

Many PCD regulatory proteins in *Arabidopsis thaliana* have been described, but one of the best known is LESION SIMULATING DISEASE (LSD). LSD encodes a small C2C2 zinc finger protein that is a negative regulator of PCD, playing the molecular function of a transcriptional regulator and scaffold protein [9]. The role of LSD1 as an important cell death regulator in *Arabidopsis thaliana* since its discovery over two decades ago has been studied intensively within both biotic and abiotic stress responses, as well as concerning plant fitness regulation. Czarnocka et al. [10] found that LSD1 plays a dual role within the cell by acting as a condition-dependent scaffold protein and as a transcription regulator. LSD1 also regulates the number and size of leaf mesophyll cells and affects plant vegetative growth. Importantly, previous research has also revealed that in addition to its function as a scaffold protein, LSD1 acts as a transcriptional regulator. The initial LSD gene, denoted as *AtLSD1*, was isolated from Arabidopsis and identified as a negative modulator of plant PCD. Studies on Arabidopsis lsd1 mutants have demonstrated their heightened sensitivity to cell death initiators, leading to an uncontrolled extent of cell death [11]. ROS has been identified as an essential and sufficient signal for propagating cell death. Hence, *LSD1* serves as a monitor for superoxide-dependent signals and exerts negative regulatory control over plant cell death pathways. Its transcriptional regulation capabilities come into play either by repressing pro-death pathways or activating anti-death pathways in response to signals originating from cells undergoing pathogen-induced hypersensitive cell death. Bernacki et al. [12] found that salicylic acid (SA) is crucial in the LSD1-dependent regulation of PCD and that the deregulation of SA synthesis or metabolism inhibits the cell death phenotype in the LSD1 background in response to abiotic stress. Nonetheless, the precise molecular regulatory mechanisms governing seed aging, particularly in relation to the *LSD* gene’s regulation of PCD and its impact on seed vigor, remain inadequately elucidated. Numerous studies have thus far failed to fully explicate the essence of seed aging and the intricate interplay between the LSD gene and PCD in this context. Further investigation is required to unravel the underlying molecular intricacies and shed light on the comprehensive regulatory network that governs seed aging and vigor.

A research report highlights the identification of four *LSD* genes in Arabidopsis, namely *AtLSD1*, *AtLOL1*, *AtLOL2*, and *AtLOL6* [9], as well as six *LSD* genes (*OsLSD1*, *OsLOL1*, *OsLOL2*, *OsLOL3*, *OsLOL4*, and *OsLOL5*), containing the zf-LSD1 structural domain in rice (*Oryza sativa*) [13]. However, a comprehensive genome-wide analysis of *LSD* genes in alfalfa (*Medicago sativa*), an agriculturally significant species extensively cultivated across 32 million hectares in North America, Europe, and Oceania [14], is yet to be conducted. In China, alfalfa is extensively cultivated across approximately four million hectares in the arid regions of the north [15]. Unfortunately, the vigor of alfalfa seeds is prone to decline during storage, leading to reduced seed vitality, the emergence of delayed and uneven seedlings in the field, and consequent declines in hay yield. Hence, the production of high-quality alfalfa seeds assumes paramount importance for efficient hay production. The recent release of the Zhongmu No. 1 alfalfa genome [16] in 2020 facilitates the comprehensive screening of the entire *LSD* gene family. Given the pivotal roles of LSD proteins in plants and the absence of prior *LSD* gene family information in alfalfa, the primary objective of this study is to identify the members of the *LSD* gene that may potentially regulate seed vigor. This study performs chromosome mapping, phylogenetic analysis, conserved domain analysis, cis-regulatory element analysis, subcellular localization prediction, and tissue-specific expression analysis of the identified *LSD* gene family members. This all-encompassing analysis of the *LSD* genes in alfalfa is expected to significantly contribute to the genetic enhancement of seed vigor in this crucial crop. Ultimately, the findings from this research endeavor could aid in developing improved alfalfa varieties with enhanced seed vigor and resilience to various stressors.

## 2. Materials and Methods

### 2.1. Identification of the MsLSD Gene Family in Alfalfa

To comprehensively identify the *MsLSD* gene family in alfalfa, the genomic information [16] of the Zhongmu No. 1 alfalfa variety was retrieved from the figshare website (https://figshare.com/articles/dataset/Medicago_sativa_genome_and_annottion_files/ accessed on 11 March 2023). To enhance the precision of this identification process, the LSD protein sequences of Arabidopsis were obtained from the PlanetTFDB database website (http://planttfdb.gao-lab.org/ accessed on 10 May 2023) and used as queries to search for possible LSD proteins within the alfalfa genome using Blast P with a stringent E-value cutoff of 1.0 × 10^−10^. Furthermore, the identified LSD protein sequences in alfalfa were subjected to multiple sequence alignment using the HMM model of HMMER3.0 (http://hmmer.org/ accessed on 11 May 2023) with an E-value of 0.001. Redundant sequences were removed from the alignment to ensure accuracy. The Hidden Markov Model (HMM) profile (PF06943) from the Pfam database (http://pfam.xfam.org/ accessed on 11 March 2023) [17] was utilized to verify the presence of the putative LSD domain in the identified sequences. Nonredundant *MsLSD* genes were designated as *MsLSD1* to *MsLSD6*, corresponding to their order of appearance in the genome file. The physical and chemical properties of the MsLSD proteins were analyzed using the ProtParam Tool (https://web.expasy.org/protparam/ accessed on 11 March 2023), with various parameters being calculated. Additionally, subcellular localization was predicted using the WoLF-PSORT tool (https://www.genscript.com/wolf-psort.html/ accessed on 6 March 2023).

### 2.2. Phylogenetic, Gene Structure, and Conserved Motif Analyses of the MsLSD Members

To explore the evolutionary relationships of *MsLSD* genes in alfalfa, a multiple sequence alignment was performed using Clustal W (http://www.clustal.org/clustal2/ accessed on 11 February 2023). An unrooted phylogenetic tree was then constructed. This was achieved using the online tool Interactive Tree Of Life (iTOL) v5 [18] and the neighbor-joining (NJ) method with 1000 bootstrap replicates. In addition to the NJ method, other phylogenetic tree construction methods were employed to ensure robustness and reliability in the tree topology [19]. To classify the *MsLSD* genes based on their evolutionary relationships with Arabidopsis *LSD* genes, a thorough analysis of the phylogenetic tree was carried out. Furthermore, conserved motifs in the MsLSD proteins were identified using the MEME tool (http://meme-suite.org/tools/meme/ accessed on 11 February 2023) [20].

### 2.3. Gene Duplication and Chromosomal Mapping Analysis

To examine the distribution of *MsLSD* genes in alfalfa chromosomal, we initially obtained the physical location information from the gene annotation file of the Zhongmu No. 1 alfalfa genome database. Subsequently, we retrieved the chromosomal locations of the *LSD* genes from the genomic annotation file GFF3. Employing the TBtools software v2.003 [21], we generated a chromosomal distribution image depicting the arrangement of *MsLSD* genes on different chromosomes. As part of the nomenclature process, the *MsLSD* genes were renamed based on their respective chromosomal locations. To explore potential gene duplication events in the *MsLSD* gene family, we employed multiple collinear scanning toolkits (MCScanX) [22] with an E-value threshold of 10^−5^. Furthermore, we conducted synteny analysis of the *MsLSD* genes between the Zhongmu No. 1 alfalfa and other plant species, including *Medicago truncatula*, *Lotus japonicus*, *Cicer arietinum*, *A. thaliana*, *Lupinus albus*, and *Glycine max*.

To visualize the exon–intron structures of the *MsLSD* genes, we employed the TBtools software, a powerful tool developed by Chen et al. [21], which allowed us to draw informative schematic diagrams based on the coding sequences (CDSs) and the corresponding full-length sequences of the identified *MsLSD* genes. This visualization enabled a comprehensive examination of the gene structures, providing valuable insights into the organization and arrangement of exons and introns within the *MsLSD* genes in alfalfa.

### 2.4. cis-Regulatory Element Analysis

To investigate the functional roles of *MsLSD* genes in response to seed aging stress in plants, we aimed to conduct a detailed analysis of the cis-regulatory elements within these genes. The promoter sequences of *MsLSD* genes were analyzed for the identification of cis-regulatory elements using the PlantCARE database (http://bioinformatics.psb.ugent.be/webtools/plantcare/html/ accessed on 11 March 2023) [23].

### 2.5. Tissue-Specific Expression Analysis of MsLSD Genes in Alfalfa

Characterizing tissue-specific expression patterns of *MsLSD* genes in alfalfa is pivotal to deciphering their functional roles within specific plant tissues. The RNA-Seq data of different tissues were downloaded from the Noble Research Institute database (https://www.alfalfatoolbox.org/ accessed on 9 March 2023). Additionally, RNA-Seq data from previous experiments involving aged seed imbibition at time points of 6, 12, 24, and 36 h were utilized to augment genetic information concerning the response of *MsLSD* genes to seed aging. These datasets were used to investigate the expression profiles of *MsLSD* genes across various tissues, and the expression patterns of all genes were visualized using TBtools software [21].

### 2.6. Plant Materials and Seed Aging Treatments

Alfalfa (cv Zhongmu No. 1) seeds were harvested in 2019 at the Forage Seed Propagation Station of China Agricultural University, situated at coordinates 37 N, 98.30 E, and an elevation of 1480 m, in Jiuquan City, Gansu province, China. Seed aging treatment was carried out following the established methodology outlined in the study by Xia et al. [24]. Briefly, seeds pre-adjusted to 10% moisture content on a fresh-weight basis were immediately sealed in an aluminum foil bag (0.12 × 0.17 m^2^, approx. 40 g in each bag) at 45 °C in a water bath. After 0 d of treatment, samples began to be taken out every four days. The germination percentage declined to 70% on the 10th day and decreased to 0% on the 32nd day (Figure 2). Seeds treated from the 0th day to the 32nd day with a range of germination percentages were obtained for subsequent tests; they were marked as D0, D4, D8, D12, D16, D20, D24, D28, and D32, respectively. The control seeds were marked as D0. Both control (CK) and aged seed samples were collected at specific time points after imbibition, specifically at 6, 12, 24, and 36 h. Each sample comprised three biological replicates, and each replicate consisted of seeds collected from at least ten individual seeds. Following the collection, the samples were rapidly frozen and ground to a fine powder using liquid nitrogen to preserve their molecular integrity.

### 2.7. RT-qPCR Analysis

Based on RNA-Seq data, some *MsLSD* genes, which exhibited a significant induction in response to seed aging treatment, were selected to validate the credibility of RNA-Seq data. The total RNA of alfalfa seeds was extracted using the RNA extraction Kit (Huayueyang Biotech Co., Ltd., Beijing, China). Subsequently, the assay of inverse transcription was conducted using the SuperMix for qPCR Kit (TransGen Biotech, Beijing, China). The qRT-PCR was conducted on a CFX96 Real-Time System. A cycle program employed for qRT-PCR consisted of an initial step at 95 °C for 3 min, followed by 40 cycles of 95 °C for 15 s and 60 °C for 30 s. The expression levels of *MsLSD* genes were calculated using the 2^−ΔΔCt^ method [25]. The results were visualized using GraphPad Prism version 8.0 (https://www.graphpad.com/ accessed on 11 March 2023). The details of the primers used for each *MsLSD* gene are listed in Appendix A. The *Medicago actin* gene was selected as the reference gene [26]. Three biological replicates for the one-time treatment of each group were run, and each reaction was performed with three technical replicates.

## 3. Results

### 3.1. Identification and Multiple Sequence Analysis of LSD Genes in Alfalfa

To investigate the distribution of *MsLSD* genes in alfalfa, we conducted a comprehensive screening of the Zhongmu No. 1 alfalfa genome. This screening employed the Hidden Markov Model (HMM) profile and BLAST searches, using 21 LSD protein sequences from *A. thaliana* and *M. truncatula* as queries. Following this process, we identified six putative MsLSD proteins, which were ultimately confirmed using the Pfam database (http://pfam.xfam.org/ accessed on 11 September 2022) based on the presence of the conserved LSD domain (PF06943). These six proteins, designated as MsLSD1 to MsLSD6, were selected for further analysis (Appendix A). To discern the chromosomal distribution of the identified *MsLSD* genes, we mapped them onto three specific alfalfa chromosomes, namely Chr2, Chr4, and Chr8 (Figure 3A). Among these chromosomes, Chr8 harbored one *MsLSD* gene, representing 16.7% of the total *MsLSD* genes. Conversely, most *LSD* genes were located on Chr2 (n = 2, accounting for 33.3%) and Chr4 (n = 3, comprising 50%). Additionally, we conducted a thorough examination of potential gene duplication events, revealing one segmental duplication event involving two *MsLSD* genes situated on chromosomes 2 and 4 (Figure 3B). These findings shed light on the genomic organization and evolutionary relationships of the *MsLSD* genes in alfalfa, providing essential insights into their potential functional significance in seed aging stress responses and plant development.

To further investigate the phylogenetic relationships within the *LSD* gene family, a collinearity analysis was conducted on the *MsLSD* genes along with those from several other plant species, including *M. truncatula*, *Lotus japonicus*, *Cicer arietinum*, *A. thaliana*, *Lupinus albus*, and *Glycine max* (Figure 4). This analysis revealed multiple homologous gene pairs between *M. sativa* and each of the aforementioned species, providing insights into their evolutionary relatedness. Specifically, *M. sativa* exhibited homologous gene pairs with *M. truncatula* (6 pairs), *L. japonicus* (6 pairs), *C. arietinum* (5 pairs), *A. thaliana* (1 pair), *L. albus* (10 pairs), and *G. max* (9 pairs). The highest homology was observed between *M. sativa* and *M. truncatula*, with the homologous pairs primarily distributed on chromosomes 2 and 8. Following this, *M. sativa* exhibited the second-highest homology with *G. max*, wherein, apart from chromosome 2, the homologous gene pairs were mainly distributed on chromosomes 8 and 4.

Sequence analyses of the MsLSD proteins unveiled considerable length variation, ranging from 69 amino acids (aa) for MsLSD1 to 502 aa for MsLSD5. The molecular weights of these proteins spanned from 7.58 kD (MsLSD1) to 57.73 kD (MsLSD5), while the isoelectric points (pI) displayed a range from 5.33 (MsLSD5) to 9.22 (MsLSD3). Notably, subcellular localization prediction results indicated that two MsLSD proteins were localized in the nucleus, two in the cytoplasm, and two in the chloroplast (Table 1). These findings collectively suggest that MsLSD proteins may exhibit diverse subcellular distribution, allowing them to potentially function in various microcellular environments.

### 3.2. Phylogenetic Analysis, Gene Structures, and Motif Composition of MsLSD Genes

To comprehensively elucidate the evolutionary history of the *LSD* gene family in alfalfa, we conducted a comparative analysis of 6 *MsLSD* genes with 59 *LSD* genes from other species, including *M. truncatula*, *A. thaliana*, *Zea mays*, and *G. max* (http://planttfdb.gao-lab.org/index.php/, accessed on 5 March 2023). To infer the evolutionary relationships, an unrooted phylogenetic tree was constructed using the neighbor-joining method in MEGAX. As depicted in Figure 5, all the identified MsLSD proteins were classified into four distinct groups, with the six LSD proteins further being grouped into two subfamilies.

To investigate the diversity of MsLSD protein motifs, a conserved motif analysis of the MsLSD protein sequences was performed using the online server MEME. This analysis revealed a total of eight distinctive conserved motifs, which were designated as motif 1 to motif 8. Notably, motif 1 and motif 7 were present in nearly all LSD genes, suggesting their conserved functional importance across the gene family (Figure 6B). Furthermore, we analyzed the exon–intron structures of the *LSD* genes to gain insights into their functional diversification. The intron–exon structure analysis demonstrated that most *MsLSD* genes possessed a limited number of introns, except for *MsLSD2* and *MsLSD6*, which contained three introns, and *MsLSD1*, which had one intron (Figure 6D). In general, the majority of *MsLSD* genes exhibited similar gene structural components.

### 3.3. cis-Regulatory Element Analysis of MsLSD Gene Promoters

To obtain a comprehensive understanding of the regulatory mechanism governing the alfalfa *LSD* gene family, we carried out a meticulous analysis of the promoter sequences of the *MsLSD* genes using the PlantCARE program. The details of six *MsLSD* gene promoter sequences are listed in Appendix A. This in-depth investigation resulted in the identification of twelve discrete types of cis-regulatory elements present within the promoter regions of the six alfalfa *LSD* gene family members. The identified cis-regulatory elements exhibited a diverse array of functional categories, encompassing pivotal aspects such as light responsiveness, MeJA responsiveness, anaerobic induction, meristem expression, abscisic acid responsiveness, salicylic acid responsiveness, gibberellin responsiveness, low-temperature responsiveness, drought inducibility, defense and stress responsiveness, seed-specific regulation, as well as auxin responsiveness (Figure 7). Among the identified cis-regulatory elements, those associated with light responsiveness were the most prevalent, followed by elements related to salicylic acid responsiveness, abscisic acid responsiveness, drought inducibility, and anaerobic induction. Moreover, all *MsLSD* genes possessed the CAAT-box and TATA-box cis-regulatory elements, indicating their widespread involvement in transcriptional regulation.

### 3.4. Secondary Structure Analysis of MsLSD Proteins

Investigating the protein’s secondary structure is paramount for gaining insights into protein function. In this study, we conducted an in-depth analysis of the secondary structure of all MsLSD proteins. In all MsLSD proteins, random coil accounted for the largest proportion (27.54–54.69%), followed by α-helix (17.69–36.06%), extended strand (17.19–42.03%), and β-turn (3.78–10.14%) (Table 2).

### 3.5. Expression Pattern Analysis of MsLSD Genes in Alfalfa Tissues

The meticulous investigation of tissue-specific expression patterns holds utmost significance in gaining profound insights into the precise functions of *MsLSD* genes across diverse tissues of alfalfa. RNA-Seq data, downloaded from the Noble Research Institute database (https://www.alfalfatoolbox.org/, accessed on 5 March 2023), were used to evaluate the transcript abundance profiles of *MsLSD* encoding genes across seven tissues, namely leaves, flowers, pre-elongated stems, elongated stems, roots, nodules, and seed imbibition at different time (6, 12, 24, and 36 h). The fpkm values of six *MsLSD* genes in different tissues are listed in Appendix A. To visually represent these expression patterns, a heatmap of expression profiles was constructed using TBtools software (Figure 8). Based on the observed expression patterns across the seven tissues of alfalfa, it was evident that *MsLSD* exhibited distinct transcript abundance in different tissues. Specifically, *MsLSD6*, *MsLSD3*, and *MsLSD2* displayed the highest transcript accumulation in elongated stems, leaves, and flowers, respectively. On the other hand, *MsLSD4* and *MsLSD5* exhibited the highest transcript accumulation in the nodule (Figure 8A). Moreover, the expression patterns of the six *MsLSD* genes also demonstrated regular changes in seeds with different vigor levels. *MsLSD2*, *MsLSD3*, and *MsLSD6* displayed the highest transcript accumulation in seeds with high vigor, while MsLSD4 and MsLSD5 exhibited the highest transcript accumulation in seeds with low vigor (Figure 8B).

To explore potential candidate genes that might regulate seed vigor, the expression pattern of the *MsLSD* genes was further analyzed during seed germination using transcriptome data (Figure 8C). Notably, *MsLSD4* and *MsLSD6* displayed high expression levels at the early seed germination stage, suggesting they might serve as positive regulators of seed vigor. However, the expression levels of *MsLSD1* and *MsLSD2* declined significantly after aged seeds had been imbibed for 36 h. Furthermore, it was observed that the six *MsLSD* genes were not continuously expressed during seed germination, but rather, they exhibited a specific expression at particular time points.

### 3.6. RNA-Seq Data Validation

To reinforce the credibility and robustness of our RNA-Seq data, we performed the RT-qPCR assay on six *MsLSD* genes. The comprehensive findings illustrated that all the targeted *MsLSD* genes displayed induction patterns during seed germination, and, notably, the expression trends were in congruence with the observations derived from the RNA-Seq analysis (Figure 9). Specifically, the expression of *MsLSD1*, *MsLSD2*, *MsLSD3*, and *MsLSD6* genes gradually decreased with the decline in seed vigor. In contrast, a noteworthy progressive increase in the expression levels of *MsLSD5* and *MsLSD6* was observed with the decline in seed vigor, thus signifying their substantial upregulation in response to the aging stress encountered by alfalfa seeds (Figure 9A–F). Notably, *MsLSD1*, *MsLSD2*, and *MsLSD3* genes showed significantly (*p* < 0.05) up-regulated expression, but *MsLSD4*, *MsLSD5*, and *MsLSD6* genes significantly (*p* < 0.05) down-regulated during 6 and 12 h of aged seed imbibition. The expression of *MsLSD1*, *MsLSD3*, *MsLSD4*, and *MsLSD6* genes markedly (*p* < 0.05) decreased after 36 h of aged seed imbibition, indicating their roles as negative regulators in seed vigor (Figure 9G–L). In summary, the expression patterns of the majority of *MsLSD* genes observed in the RT-qPCR assay were in concordance with those identified in the RNA-Seq analysis. Nonetheless, it is worth noting that there were variations in the fold change between the RT-qPCR and RNA-Seq data. These findings serve to provide additional validation for the credibility of our RNA-Seq data and lend support to the resilience of the differential expression patterns exhibited by *MsLSD* genes in response to the stress induced by seed aging.

## 4. Discussion

PCD is the ultimate end of the cell cycle, which occurs in all living multicellular organisms. Some of the best-described Arabidopsis proteins in the context of PCD regulation are *LSD1*, *EDS1*, and *PAD4* [27,28]. *LSD1* is a negative regulator of PCD, suppressing EDS1 and PAD4 activities since the double mutants *eds1/lsd1* and *pad4/lsd1* demonstrate a reverted *lsd1*-specific phenotype in terms of ethylene and ROS accumulation and cell death [27]. LSD, a member of the zinc finger protein family, serves as a pivotal regulator of PCD transcription factors [9] and plays a vital role in the regulation of plant development among flowering plants [29]. Previous studies have extensively conducted whole-genome identification, evolutionary relationship analysis, and molecular function investigation of LSD TFs in model plants such as *A. thaliana* [11] and *O. sativa* [13]. In this investigation, a comprehensive analysis of the *MsLSD* gene family was conducted in the Zhongmu No. 1 alfalfa genome, resulting in the identification of six distinct *MsLSD* genes that were unevenly mapped on three alfalfa chromosomes. The total number of LSD genes in alfalfa was found to be lower than those identified in other plant species, such as *M. truncatula* (9) [30], *A. thaliana* (12) [31], *Z. mays* (20) [32], and *G. max* (18) [33]. Notably, the distribution of *MsLSD* genes on chromosomes revealed gene duplication events, which were significant drivers of species evolution, contributing to the generation of novel gene functions and the expansion of gene families. These *MsLSD* genes exhibited variation in molecular weights, amino acid sequence lengths, and theoretical isoelectric points, indicating the complexity of the alfalfa genome and the functional divergence of *MsLSD* genes in response to diverse biotic and abiotic stresses. Furthermore, phylogenetic analysis classified the *MsLSD* genes into four subgroups, with the neighbor-joining (NJ) method yielding results more closely aligned with those observed in Arabidopsis and rice. This phylogenetic analysis sheds light on the evolutionary relationships among LSD genes in different plant species, providing valuable insights into the conservation and diversification of this gene family across plants.

TFs play a pivotal role in the precise regulation of gene expression by selectively binding to specific cis-acting elements located in the promoter regions of target genes, which often function as families, enabling rapid responses to environmental changes and regulating various biological processes [34]. The LSD TFs, including *AtLSD1* in Arabidopsis and *OsLSD1* in rice, have been shown to participate in diverse regulatory pathways [11,35]. For instance, the *OsLSD1* gene in rice encodes a zinc finger structural protein that positively regulates the resistance pathway to blast fungus [13]. *OsLSD1* is also involved in light- and temperature-dependent metabolic pathways, with changes in light and temperature leading to leaf deformation in *Oslsd1* mutants, affecting rice growth and development [13]. The diverse array of regulatory functions exhibited by TFs stems from the presence of various cis-acting sites. Previous studies provided evidence of the functional diversity of LSD proteins, implicating their significant roles in regulating plant growth and stress responses. This notion has demonstrated the prevalence of numerous stress-responsive elements within the promoter regions of *LSD* genes in Arabidopsis and rice [11,13]. In our study, we successfully identified an abundant array of cis-regulatory elements related to hormones and stress in the promoter regions of the identified *MsLSD* genes. Notably, we frequently detected the promoter elements associated with hormone responses, such as MeJA responsiveness, abscisic acid responsiveness, salicylic acid responsiveness, gibberellin responsiveness, auxin responsiveness, and gibberellin-responsive elements. Furthermore, our observation of seed-specific regulatory elements highlights the potential significance of *MsLSD* genes in seed development and germination. In light of our findings that the upstream promoters of the alfalfa *LSD* gene family contain numerous photo-responsive elements (Figure 5), we speculate that the LSD proteins in alfalfa may participate in three regulatory pathways. First, they may be involved in alfalfa’s response to environmental stress. Second, they could play a role in the regulation of chlorophyll synthesis and chloroplast development. Lastly, LSD proteins may be involved in the formation of aerenchyma in alfalfa and the absorption and utilization of nitrogen, thus influencing the growth and development of the plant. Bernacki et al. [27] found that the LSD1 proteins are conditional molecular regulators of seed yield through SA and H_2_O_2_ signaling. Our study offers significant contributions in terms of valuable insights into the potential functions of *MsLSD* genes, both within diverse tissues and under varying seed vigor conditions. Moreover, it sheds light on the possible regulatory pathways in which the LSD proteins in alfalfa may be involved.

The gene expression profiles present invaluable insights and are pivotal indicators for predicting the potential functions of *MsLSD* genes [36]. An in-depth analysis of tissue-specific expression patterns assumes utmost importance in unraveling the specific roles of *MsLSD* genes across diverse tissues of alfalfa. Our analysis revealed distinct transcript abundance patterns for *MsLSD* genes across different tissues. Specifically, *MsLSD6*, *MsLSD3*, and *MsLSD2* exhibited the highest transcript accumulation in elongated stems, leaves, and flowers, respectively. Conversely, *MsLSD4* and *MsLSD5* showed the highest transcript accumulation in nodules. Furthermore, the expression patterns of the six *MsLSD* genes displayed regular changes in seeds with different vigor. Notably, *MsLSD2*, *MsLSD3*, and *MsLSD6* showed the highest transcript accumulation in high-vigor seeds, while *MsLSD4* and *MsLSD5* showed the highest transcript accumulation in low-vigor seeds. To gain further insights into the expression patterns and putative functions of alfalfa *LSD* genes, we selected six *MsLSD* genes and examined their expressions in response to seed aging stress. The data supported the expression profiles of *MsLSD* genes in response to different biotic and abiotic stresses, providing additional evidence for the regulatory roles of *MsLSD* genes under stress conditions. To ensure the reliability of our RNA-Seq data, we performed RT-qPCR validation for six *MsLSD* genes (*MsLSD1*, *2*, *3*, *4*, *5*, and *6*). The expression level of *MsLSD1*, *MsLSD2*, *MsLSD3*, and *MsLSD6* gradually decreased as seed vigor declined, suggesting their negative regulatory roles in seed aging. However, the expressions of *MsLSD4* and *MsLSD5* increased gradually as seed vigor declined, indicating that they play positive regulatory roles in seed aging. Moreover, the expression level of *MsLSD5* and *MsLSD6* exhibited significant (*p* < 0.05) up-regulation in aged seed imbibition for 6 h, as well as a heightened sensitivity to the aging treatment. Conversely, *MsLSD1*, *MsLSD3*, and *MsLSD4* manifested analogous variations in both CK and aged seeds, demonstrating substantial alterations solely during the initial imbibition phase, while their responsiveness to seed aging exhibited a delayed response. In contrast to the expression profiles of the remaining genes, *MsLSD2* exhibited significant (*p* < 0.05) down-regulation in aged seed imbibition for 6, 12, and 36 h. This distinctive pattern of expression indicates the active involvement of *MsLSD2* in the negative regulation of seed vigor, potentially signifying the pivotal role it plays within the *MsLSD* gene family.

Many studies have found that seed aging is closely related to PCD, and a quarter of a century ago, LSD1, EDS1, and PAD4 were discovered as regulators of PCD. Bernacki et al. [37] focus on the specific features of LSD1, EDS1, and PAD4 that make them potentially important for agricultural and industrial use. However, the intricate regulatory mechanisms governing seed vigor related to these *LSD* genes in alfalfa still necessitate further elucidation. Thus, further investigations on individual *MsLSD* genes are warranted in the future. For instance, *MsLSD2* may contribute to oxidative stress tolerance during alfalfa seed storage, *MsLSD5* might play a crucial role in abscisic acid signal transduction, and *MsLSD6* could be involved in the DNA repair pathway during alfalfa seed germination. Unraveling the specific functions of these individual *MsLSD* genes will further our understanding of the molecular mechanisms underlying seed vigor in alfalfa.

## 5. Conclusions

In conclusion, this study successfully identified six *MsLSD* genes and mapped them onto three chromosomes in alfalfa. Phylogenetic analysis categorized these genes into four subfamilies, and subcellular localization prediction indicated that most MsLSD proteins were localized in the nucleus. Gene structure analysis revealed that there was high conservation among *MsLSD* genes, with most of them having very few or no introns. Among the six LSD TFs, three genes showed significant induction in response to seed aging, suggesting they might play an important role in maintaining seed vigor. Remarkably, the prominent responses observed in *MsLSD2* and *MsLSD6*, particularly under aging stress, suggest their potential significance in inhibiting programmed cell death during seed imbibition under unfavorable conditions. These comprehensive findings have shed light on the molecular basis of *LSD* gene families. However, the intricate regulatory mechanisms governing seed vigor in relation to these *LSD* genes in alfalfa still necessitate further elucidation. Thus, further investigations into individual *MsLSD* genes are warranted in the future. Unraveling the specific functions of these individual *MsLSD* genes will deepen our understanding of the molecular mechanisms underlying seed vigor in alfalfa. The present study will lay the foundation for future studies on verifying the precise regulation of *LSD* genes in *M*. *sativa*. Finally, aging stress-responsive *MsLSD* genes can be used to bring about a genetic improvement in the stress resistance and seed vigor of alfalfa.

## Figures and Tables

**Figure 1 antioxidants-12-01768-f001:**
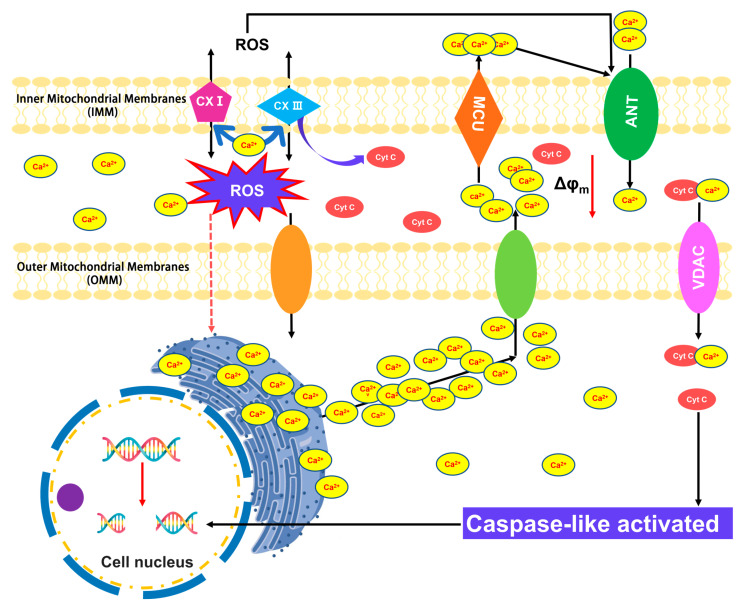
Schematic representation of ROS-dependent signaling involved in programmed cell death (PCD). Increased reactive oxygen species (ROS) production from mitochondria may trigger Ca^2+^ influx from the endoplasmic reticulum (ER), and cytoplasm enters the mitochondrial matrix through voltage-dependent anion channels (VDACs) or calcium uniporter (MCU). High levels of Ca^2+^ in the mitochondrial stimulate respiratory chain (CXI and CXIII) activity, which leads to higher amounts of ROS, which can further lead to the increased release of Ca^2+^. Further, the increased ROS and Ca^2+^ load in mitochondria will promote the opening of mitochondrial permeability transition pores (MPTPs), which span the inner (IMM) and outer mitochondrial membranes (OMM) and are mainly composed of cyclophilin D (D), adenine nucleotide translocator (ANT), and VDAC. The prolonged opening of MPTP leads to a significant increase in the permeability of the IMM to solutes with molecular masses < 1500 Da as a consequence of the charge difference between the mitochondrial matrix and the cytosol (mitochondrial membrane potential, Δφ_m_). Subsequently, mitochondria swell irreversibly, causing the OMM to rupture while releasing cytochrome C (Cyt C) from inter-membrane space to the cytoplasm. High Cyt C levels will initiate the activity of caspase-like enzymes, which will enter the nucleus, resulting in typical PCD changes such as DNA fragmentation and disruption of the nucleus.

**Figure 2 antioxidants-12-01768-f002:**
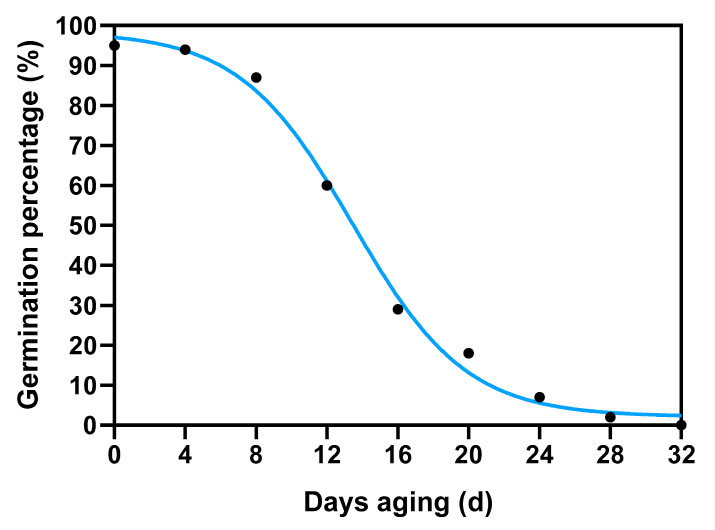
Effect of aging treatments on alfalfa seed germination.

**Figure 3 antioxidants-12-01768-f003:**
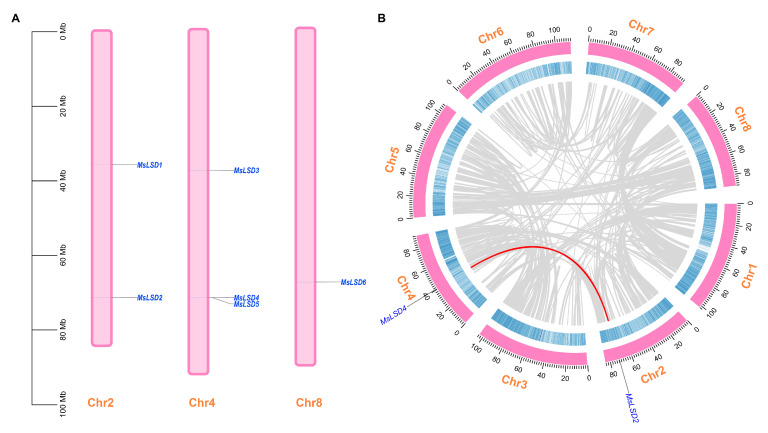
Chromosomal distribution of *MsLSD* genes (**A**), and segmentally duplication events of *MsLSD* genes (**B**). The segmentally duplicated genes are connected by red lines, referring to the 2 genes highlighted in blue.

**Figure 4 antioxidants-12-01768-f004:**
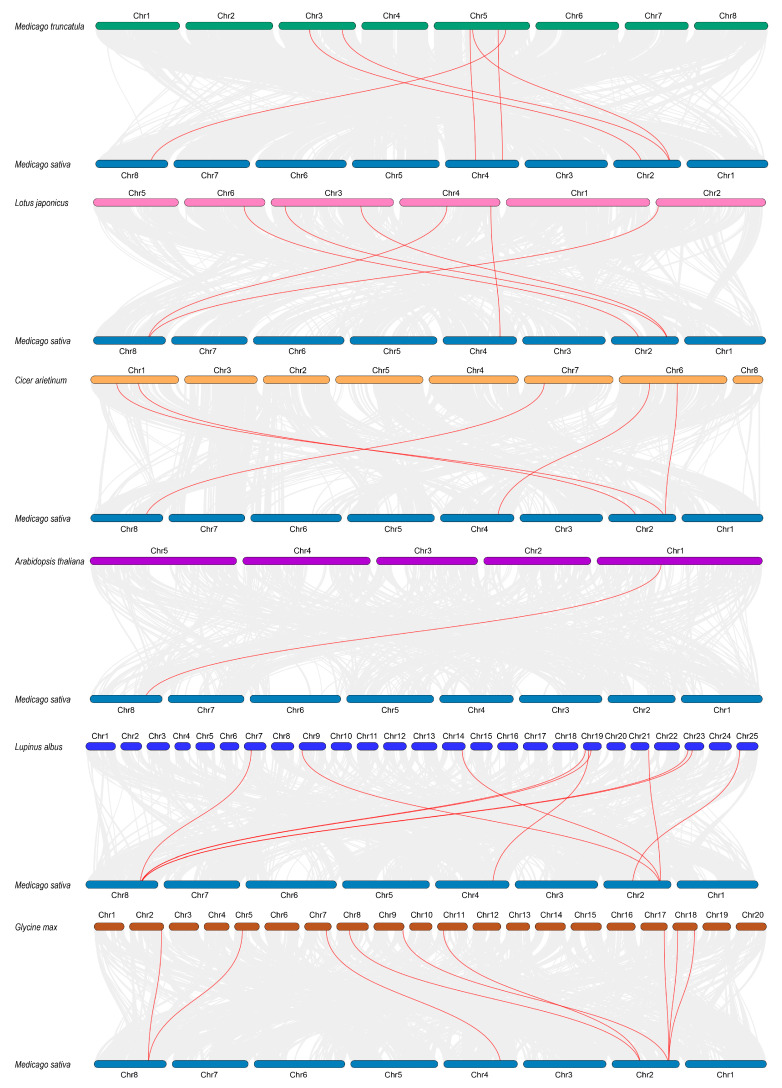
Syntenic analysis of *MsLSD* genes in *M. sativa* compared with those in six plant species (*M. truncatula*, *L. japonicus*, *C. arietinum*, *A. thaliana*, *L. albus*, and *G. max*).

**Figure 5 antioxidants-12-01768-f005:**
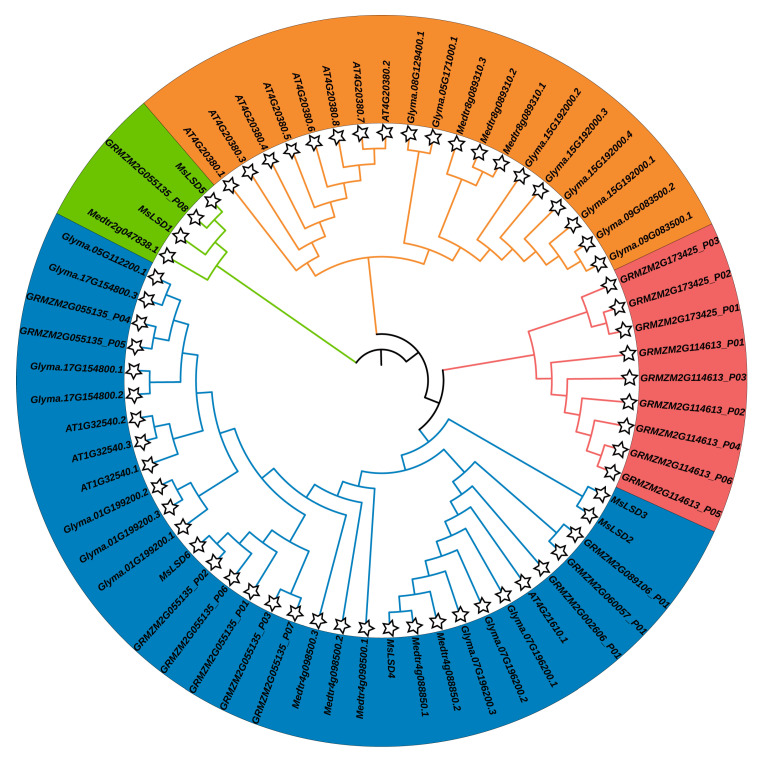
Phylogenetic analysis of LSD proteins. The phylogenetic tree was generated by the neighbor-joining method derived from Clustal X alignment of MsLSD amino acid sequences from *M. truncatula*, *A. thaliana*, *Z. mays*, *G. max*, and *M. sativa*.

**Figure 6 antioxidants-12-01768-f006:**
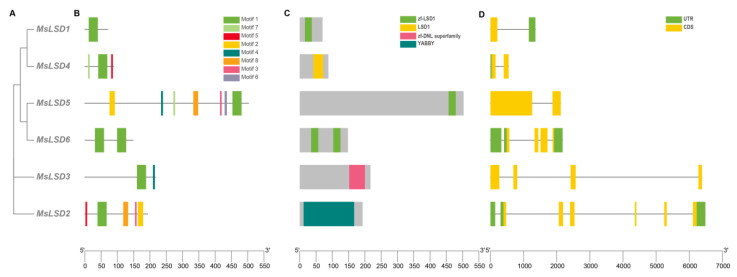
Phylogenetic tree, conserved motif, and gene structure of *MsLSD* family members. (**A**) The phylogenetic tree of *MsLSD* genes was constructed using MEGA-X and the neighbor-joining (NJ) method. (**B**) The conserved motif of MsLSD proteins was analyzed on the MEME tool, and the results were visualized using TBtools. The motifs, labeled as 1–8 and represented by different colored boxes, demonstrate the conserved patterns in the protein sequences. The scale at the bottom allows for an estimation of the protein lengths. (**C**) The conserved domain analysis of LSD protein, and different colors indicate different conserved domains. (**D**) The gene structure of *MsLSD* genes was determined, with the coding regions for the MsLSD domain being highlighted in yellow. Regions labeled as CDS without the region coding for the MsLSD domain are indicated in green.

**Figure 7 antioxidants-12-01768-f007:**
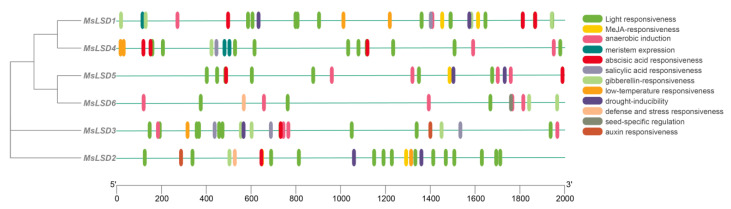
cis-regulatory elements in the promoters of *MsLSD* gene families.

**Figure 8 antioxidants-12-01768-f008:**
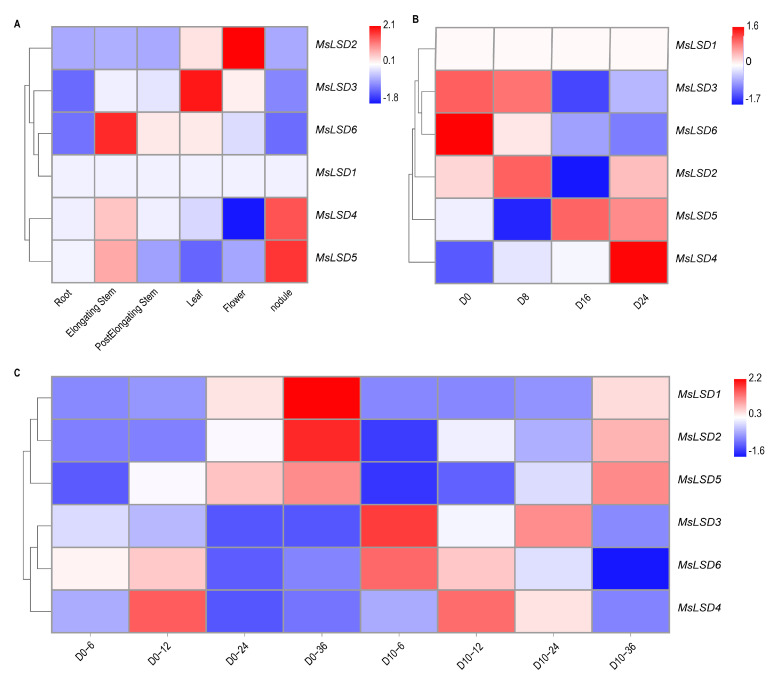
Expression pattern of 6 *MsLSD* genes in different tissues and seeds of different vigor levels based on RNA-seq data. (**A**) Expression profile of 6 *MsLSD* genes in different tissues (leaves, flowers, pre-elongated stems, elongated stems, roots, nodules, and seed) of alfalfa. (**B**) Expression patterns of the *MsLSD* gene in seeds aged 0 days (D0), 8 days (D8), 16 days (D16), and 24 days (D24) with different vigor levels. (**C**) Expression profile of 6 *LSD* genes in alfalfa seed imbibed for 6, 12, 24, and 36 h under aging treatment. The color scale represents log10 expression values. Red represents up-regulated expression, and blue represents down-regulated expression.

**Figure 9 antioxidants-12-01768-f009:**
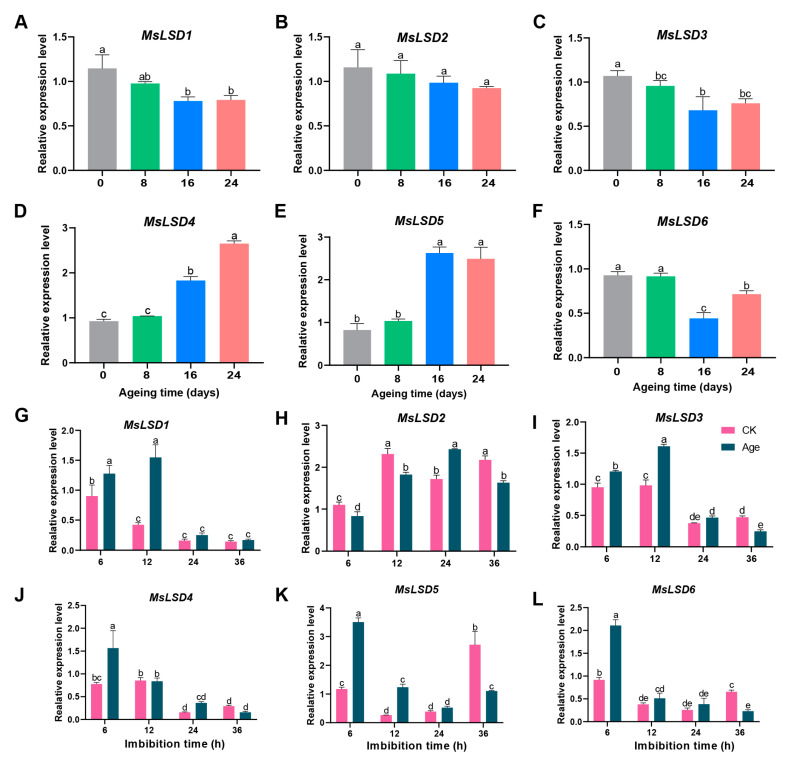
Expression profile of *MsLSD* genes via RT-qPCR assay. (**A**–**F**) Expression level of six *MsLSD* genes in different vigor alfalfa seeds; (**G**–**L**) expression level of six *MsLSD* genes under seed imbibed for 6, 12, 24, and 36 h after aging treatment using RT-qPCR assay. The data are the means of three biological replicates ± SEM. Different lowercase letters represent significant differences, and *p* < 0.05 was considered highly significant.

**Table 1 antioxidants-12-01768-t001:** Properties and predicted locations of MsLSD proteins.

Gene Name	Gene Id in ‘Zhongmu No. 1’ Assembly	Isoelectric Point	Molecular Weight (Da)	Amino Acids	Subcellular Localization	Instability Index	Average of Hydropathicity
*MsLSD1*	MsG0280008660.01. T 01	9.13	7588.24	69	Cytoplasm	56.26	1.139
*MsLSD2*	MsG0280010569.01. T 01	9.08	21,433.39	192	Nucleus	47.59	−0.543
*MsLSD3*	MsG0480020269.01. T 01	9.22	24,524.33	216	Nucleus	50.53	−0.223
*MsLSD4*	MsG0480022392.01. T 01	8.37	9638.29	87	Chloroplast	75.9	−0.333
*MsLSD5*	MsG0480022393.01. T 01	5.33	57,730.72	502	Cytoplasm	46.51	−0.237
*MsLSD6*	MsG0880046041.01. T 01	6.77	15,775.28	147	Chloroplast	39.12	0.211

**Table 2 antioxidants-12-01768-t002:** The secondary structure of MsLSD proteins.

Gene Name	Gene Id in ‘Zhongmu No. 1’ Assembly	α-Helix (%)	Extended Strand (%)	β-Turn (%)	Random Coil (%)
*MsLSD1*	MsG0280008660.01. T 01	20.29%	42.03%	10.14%	27.54%
*MsLSD2*	MsG0280010569.01. T 01	22.92%	17.19%	5.21%	54.69%
*MsLSD3*	MsG0480020269.01. T 01	24.54%	22.69%	6.94%	45.83%
*MsLSD4*	MsG0480022392.01. T 01	21.84%	21.84%	8.05%	48.28%
*MsLSD5*	MsG0480022393.01. T 01	36.06%	17.93%	3.78%	42.23%
*MsLSD6*	MsG0880046041.01. T 01	17.69%	26.53%	5.44%	50.34%

## Data Availability

Data are contained within the article and Appendix A. Raw sequencing data of the transcriptome used in the current study are available in the NCBI’s Sequence Read Archive (SRA, https://www.ncbi.nlm.nih.gov/sra, accessed on 3 September 2023) under the BioProject PRJNA1012365. The genomic information of the Zhongmu No. 1 alfalfa variety was retrieved from the figshare website (https://figshare.com/articles/dataset/Medicago_sativa_genome_and_annottion_files/, accessed on 11 March 2023). The RNA-Seq data, downloaded from the Noble Research Institute database (https://www.alfalfatoolbox.org/, accessed on 5 March 2023), were used to evaluate the transcript abundance profiles of *MsLSD* encoding genes across seven tissues, namely leaves, flowers, pre-elongated stems, elongated stems, roots, and nodules.

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
