# Peer review of "Genomic Identification and Expression Profiling of Lesion Simulating Disease Genes in Alfalfa (Medicago sativa) Elucidate Their Responsiveness to Seed Vigor"

_antioxidants, 2023, doi:10.3390/antiox12091768_

Round 1

Reviewer 1 Report

The manuscript entitled "Genomic Identification and Expression Profiling of LSD Genes 2 in Alfalfa (Medicago sativa) Elucidate Their Responsiveness to 3 Seed Vigor" is excellent and can be acceptable after some revisions.

1) The authors should describe how the aging treatment was done in the materials and methods more precisely. The authors only refer to the reference by Xia et al. But the aging treatment is very important for this study, please add the method.  

2) Please add the data on the percentage of germination with aging treatment and the control seeds. We cannot see how the aging treatment gave harmful effects on the seed germination of alfalfa.

3) The legends of Figure 7 are not clear.

Are these data the relative expression of LSDs against reference genes?

Figure 7 A: What date did you analyze the expression of LSDs among organs? 

Figure 7C: Did you compare the D0 not aged, and D10 aged?

Author Response

Thank you very much for taking time out of your busy schedule to review our work.

Thank you for your careful review and professional comments and questions.

1) The authors should describe how the aging treatment was done in the materials and methods more precisely. The authors only refer to the reference by Xia et al. But the aging treatment is very important for this study, please add the method.  

Response: Thanks for the suggestion. We strongly agree with the reviewer that the aging treatment is very important for this study. We added the detailed method of seed aging in the method section.

2) Please add the data on the percentage of germination with aging treatment and the control seeds. We cannot see how the aging treatment gave harmful effects on the seed germination of alfalfa.

Response: Thanks for the suggestion. We have added a Figure 2 to the main manuscript method section to show the effect of aging treatments on alfalfa seed germination.

3) The legends of Figure 7 are not clear.

Are these data the relative expression of LSDs against reference genes?

Figure 7 A: What date did you analyze the expression of LSDs among organs? 

Figure 7 C: Did you compare the D0 not aged, and D10 aged?

Response: Dear reviewer, I'm so sorry that the legends of Figure 7 was not clearly written, so that you can’t understand our work.

  1. The analysis of 6 MsLSD genes expression pattern was based on RNA-seq data, and raw sequencing data of the seed transcriptome used in the current study are available in the NCBI’s Sequence Read Archive (SRA, https://www.ncbi.nlm.nih.gov/sra) under the BioProject PRJNA1012365.
  2. The RNA-Seq data, which was downloaded from Noble Research Institute database (https://www.alfalfatoolbox.org) was used to evaluate transcript abundance profiles of MsLSD encoding genes across seven tissues, namely leaves, flowers, pre-elongated stems, elongated stems, roots, nodules.
  3. Figure 7 C, we wanted to explore the expression pattern of MsLSD gene during D0 not aged, and D10 aged seed imbibition. We only analyzed the expression of MsLSD gene in seeds imbibition for 6,12,24 and 36h.

Reviewer 2 Report

The manuscript titled "Genomic Identification and Expression Profiling of LSD Genes in Alfalfa (Medicago sativa) Elucidate Their Responsiveness to Seed Vigor" by Sun et al. presents an exploration of the Lesion Simulating Disease (LSD) gene family in alfalfa, with a particular emphasis on its role in seed vigor. While this work holds promise, several key aspects require significant improvement.

Concerns:

Introduction: The introduction and discussion sections lack essential references, which are crucial for contextualizing the study. Some critical references, such as "Cells. 2021 Apr 20;10(4):962" and "Plant Cell Environ. 2017 Nov;40(11):2644-2662," have been omitted. Additionally, the molecular and functional definition of LSD genes should be clearly outlined.

Figure 1: This figure needs enhancement. It should provide a more detailed description of the processes depicted and include references for the sources of this information. The interconnections between the different processes should be made explicit. Figures should be self-explanatory to facilitate understanding.

Table S3: The gene names in Table S3 are not clear. It appears that "WRKY" is used as a gene id, which should be clarified or corrected.

Critical Points:

Materials and Methods (M&M): The Materials and Methods section requires more detailed information. Specifically, a housekeeping gene for Medicago sativa in the studied tissues and physiological processes, particularly seed imbibition, needs to be defined. Additionally, primer sequences for reference genes should be included in the primer list.

Database Accessibility: Access to the databases used for obtaining the Medicago sativa genome and RNAseq data is essential. Functional links should be provided, as this is crucial for the review process and verifying the results.

Lack of Sequence Information: The manuscript mentions the analysis of LSD gene promoters but does not provide the actual sequence information. This omission should be addressed.

Tetraploid Nature of Medicago sativa: Authors should address how they dealt with the fact that Medicago sativa is a tetraploid species, as this can impact gene expression and interpretation of results.

Concluding Remarks: To assert that MsLSD2, MsLSD5, and MsLSD6 are involved in seed aging in alfalfa, the authors should consider incorporating additional biological evidence beyond expression analysis. Revisiting the paper's conclusion and its overall focus is advisable.

In summary, while the manuscript shows promise, addressing these concerns and critical points is absolutely essential.

The English is generally fine; only minor corrections are required.

Author Response

Thank you very much for taking time out of your busy schedule to review our work.

Thank you for your careful review and professional comments and questions.

Concerns:

Introduction: The introduction and discussion sections lack essential references, which are crucial for contextualizing the study. Some critical references, such as "Cells. 2021 Apr 20;10(4):962" and "Plant Cell Environ. 2017 Nov;40(11):2644-2662," have been omitted. Additionally, the molecular and functional definition of LSD genes should be clearly outlined.

Response: Thank you for your kind and responsible suggestions. Dear reviewer, I'm so sorry that we omitted some critical references, which revealed LSD1, EDS1 and RAD4 as regulators of programmed cell death and biotic stress responses in plant. We improved introduction and discussion sections. In addition, we outlined the molecular and functional definition of LSD genes.

Figure 1: This figure needs enhancement. It should provide a more detailed description of the processes depicted and include references for the sources of this information. The interconnections between the different processes should be made explicit. Figures should be self-explanatory to facilitate understanding.

Response: Dear reviewer, thank you for your careful review and professional comments. I'm so sorry that Figure 1 is not clearly drawn, making it difficult for the author to understand. We have redrawn Figure 1 to make it clearer and easier to understand, and also provide a more detailed description of the processes depicted and include references for the sources of this information in the manuscripts.

Table S3: The gene names in Table S3 are not clear. It appears that "WRKY" is used as a gene id, which should be clarified or corrected.

Response: Dear reviewer, thank you for your careful review. I'm so sorry that we did not check the S3 information carefully. we corrected the gene names in Table S3.

Critical Points:

Materials and Methods (M&M): The Materials and Methods section requires more detailed information. Specifically, a housekeeping gene for Medicago sativa in the studied tissues and physiological processes, particularly seed imbibition, needs to be defined. Additionally, primer sequences for reference genes should be included in the primer list.

Response: Dear reviewer, thank you for your careful review. We added housekeeping gene detailed information to the methods section, and the primer information is listed in S1. The Medicago actin gene was selected as the reference gene.

Database Accessibility: Access to the databases used for obtaining the Medicago sativa genome and RNAseq data is essential. Functional links should be provided, as this is crucial for the review process and verifying the results.

Response: Dear reviewer, I'm so sorry that RNAseq data was not uploaded to NCBI database in time. Data is contained within the article and supplementary material. Raw sequencing data of the transcriptome used in the current study are available in the NCBI’s Sequence Read Archive (SRA, https://www.ncbi.nlm.nih.gov/sra, accessed on 3 September 2023) under the BioProject PRJNA1012365. The genomic information of the Zhongmu No. 1 alfalfa variety was retrieved from the figshare website (https://figshare.com/articles/dataset/Medicago_sativa_genome_and_annottion_files/ accessed on 11 March 2023). The RNA-Seq data, which was downloaded from Noble Research Institute database (https://www.alfalfatoolbox.org/accessed on 5 March 2023) was used to evaluate transcript abundance profiles of MsLSD encoding genes across seven tissues, namely leaves, flowers, pre-elongated stems, elongated stems, roots, nodules.

Lack of Sequence Information: The manuscript mentions the analysis of LSD gene promoters but does not provide the actual sequence information. This omission should be addressed.

Response: Dear reviewer, I'm so sorry that we forgot to provide a detailed sequence information of MsLSD gene promoters. We have provided the promoter sequence in the supplementary materials S3.

Tetraploid Nature of Medicago sativa: Authors should address how they dealt with the fact that Medicago sativa is a tetraploid species, as this can impact gene expression and interpretation of results.

Response: Thanks for the questions and suggestions. We strongly agree with the reviewer for tetraploid impact on gene expression. Medicago sativa is an autotetraploid species, and our expression analysis was based on the combined expression from four haplotypes, as Zhongmu NO.1 genome assembly is with one set of chromosomes. It is expected that the expression level for one specific gene should be contributed by all the homologous genes from four haplotypes, and genes’ fpkm values were comparable and calculated based on the same strategy.

Concluding Remarks: To assert that MsLSD2, MsLSD5, and MsLSD6 are involved in seed aging in alfalfa, the authors should consider incorporating additional biological evidence beyond expression analysis. Revisiting the paper's conclusion and its overall focus is advisable.

Response: Thanks for the questions and suggestions. We strongly agree with the reviewer for the only analysis of gene expression does not asserting that MsLSD2, MsLSD5, and MsLSD6 are involved in seed aging. We rearranged the focus of the paper and rewrote the conclusion. The focus of this paper is to identify LSD genes at the whole genome level. In addition, we initially analyzed the expression patterns of these genes in different vigor seeds, trying to find some key LSD genes that regulate seed vigor, and lay a foundation for the subsequent investigation of the relationship between programmed cell death and seed vigor.

Round 2

Reviewer 2 Report

Dear Authors,

I appreciate your prompt response. Overall, I am satisfied with the adjustments made in response to my previous criticisms. Figure 1 is now much more interpretable, and you have adequately addressed concerns regarding the reference gene for qPCR and the dataset.

I would like to recommend that you incorporate the definition of LSD genes into the abstract for clarity and comprehensiveness.

Additionally, in the abstract, on line 27, it currently reads "seed germinating." I believe it should be revised to "germinating seed." I recommend reviewing the entire manuscript for similar inconsistencies.

Wishing you all the best,

Minor details, as I previously mentioned in my comments to the authors.

Author Response

Dear reviewer,

Thanks for your careful review and professional comments.

I would like to recommend that you incorporate the definition of LSD genes into the abstract for clarity and comprehensiveness.

Response: Dear reviewer, I'm so sorry that we did not clearly define the LESION SIMULATING DISEASE (LSD) gene in the abstract section, because the abstract is a separate section, it should clearly define the genes we are studying.

Additionally, in the abstract, on line 27, it currently reads "seed germinating." I believe it should be revised to "germinating seed." I recommend reviewing the entire manuscript for similar inconsistencies.

Response: Dear reviewer, Thanks for your careful review and professional comments. We have revised reads "seed germinating" to "germinating seed", and reviewed the entire manuscript.

Comments on the Quality of English Language

Minor details, as I previously mentioned in my comments to the authors.

Response: Dear reviewer, MDPI's English Editor "Holli Davies" has carefully revised the language and grammar of this manuscripts.